# Transitus: The Discontinuity and Death of Religious Communities in the Twenty-First Century

## Krijn Pansters

Tilburg School of Catholic Theology, Franciscan Study Center, Nieuwegracht 61, 3512 LG Utrecht, The Netherlands; k.pansters@tiburguniversity.edu

**Abstract:** In religious community life, the question of one's own finitude in the perspective of the infinite is always at play, but communities are now really having to let go of organizational and spiritual patterns on the one hand and develop new ones on the other. Probably clearer than anywhere else in the world, the decades-long exodus of organized religious life can be seen and felt in the Netherlands and Belgium, where centuries-old congregations are now being dissolved and their monasteries repurposed. In this article, I will study the phenomenon of religious discontinuity and death from five analytical perspectives: historical, sociological, organizational, psychological, and theological. An overall perspective is that of "transitus spirituality", of individual ways (as well as inabilities) to experience a form of completion and find closure when dealing with the discomforting fact of collective discontinuity and corporate death, of the (imminent) end of one's own community. I will conclude with a reflection on the future of organized religious life.

**Keywords:** transitus; religious life; organizational death; spirituality

*Believe in Me and have faith in My mercy. When you think you are far from Me, I am often more near. When you think almost everything is lost, often the gain of greater merit dawns. All is not lost when something is in adversity. You must not judge according to your present feelings, nor so much as let a difficulty, wherever it comes from, take hold of you and conceive of it as if all hope of coming out on top had vanished. Do not think that you have been totally abandoned, even if I temporarily send you affliction or deprive you of the desired consolation. After all, this is how one passes into the kingdom of heaven.*

Thomas a Kempis, *The Imitation of Christ* IV,30

*Nihil stabile in humanis, nihil perpetuum in caducis.*

Johannes Trithemius, *Sermo* 2

## 1. Introduction

The history of the Western Church and the research on it show that religious orders typically go through epochs of emergence, growth, maturity, crisis, and renewal. A historically understudied phase in the development of religious orders is that of *aging*, of approaching the end of the organizational life cycle (see "Historical Perspective", below). This end phase, an irreversible process of waning vitality, is particularly evident today. The undeniable evidence of an autumn of religious life in the West must be seen against the background of the general ecclesiastical decline and the rapid vanishing of traditional religious practices, especially since the Second Vatican Council.[1] The transformation of culture, society, and Church in the mid-to-late twentieth-century caused crises of identity, authority, and recruitment, which saw the numbers and duties of religious orders diminished in unforeseen and unforeseeable degrees.[2] Probably clearer than anywhere else in the world, the decades-long exodus of organized religious life can be seen and felt in the Netherlands and Belgium, where centuries-old congregations are now being dissolved and their monasteries repurposed (see "The Future of Religious Life", below).

In religious community life, the question of one's own finitude in the perspective of the infinite is always at play, but communities are now really having to let go of organizational and spiritual patterns on the one hand and develop new ones on the other (see "Organizational Perspective", below). Religious communal living, although always in transition, is aimed at perpetuating its own form of life, at following a rule in the perspective of eternity.[3] When facing one's own finitude—preparing for the end—becomes an inescapable and all-pervading reality, the eye on the future and the eye on the past are both turned toward what really matters here and now: the core spiritual principles and practices that one has always held on to and must continue to hold on to.[4] What is it in our spirituality that gives us strength in this situation and that *has* to be continued, captured, conveyed, and communicated in this transition? While for some communities this question presents a negative and painful confrontation with what is really no longer there, for others, answers seem to lie in developing new forms and new focal points. At any rate, where disillusionment and disorientation have not already had devastating effects, they do remain dominating elements in the process of finding a future.

In this article, I will study the phenomenon of religious discontinuity and death from five analytical perspectives: historical, sociological, organizational, psychological, and theological. An overall perspective is that of "transitus spirituality," of individual ways (as well as inabilities) to experience a form of completion and find closure when dealing with the discomforting fact of collective discontinuity and corporate death, of the (imminent) end of one's own community.[5] I will conclude with a reflection on the future of organized religious life.

## 2. Historical Perspective

The historical literature on the development of individual orders is too vast to mention. Theories or schemes of developmental stages are foreign to the modern historical literature, which usually focuses on particularities in contexts.[6] Where structures and commonalities are mentioned, sociological theories often abound. There are roughly two predominant research perspectives. The most recent of these is comparative order history, especially as developed in the 1990s by the historian Gert Melville.[7] Typically, while this research perspective deals with institutional durability, decline, renewal, and the like, it does not (or hardly) deal with organizational aging and death. The older research perspective is that of psychosociological order history, especially as developed in the 1960s by pastoral psychologist and psychosociologist Raymond Hostie. He summarizes the institutional development of religious orders as follows:

> After a longer or shorter incubation period, this expansion reaches its peak in one or two centuries. It more or less maintains itself, until decay sets in, which seems inevitable despite the efforts of the hierarchical instances, who wish to preserve the good of their contribution, and also despite the diligence of some of the order's members, who are animated by the desire not to slacken in the zeal of former times. These efforts are not necessarily without results. In fact, we see many reforms coexisting and passing on the life. (Hostie 1972, p. 79)

> Here, attention to organizational death is indeed—a priori and conceptually—central:

> Finally, we come up against the inescapable question—this too is supported by evidence—of the death of these groups and the death struggle that precedes it.[8]

Hostie actually searched the history of Western organized spirituality for religious foundations as well as abolitions. As his table (see Table 1) shows, 276 religious orders were founded and 99 were abolished between the fourth and twentieth centuries:

**Table 1.** Overview of foundations and abolished orders throughout the ages. Detail of Table in Hostie (1972, p. 348).

| Century | Foundation | Extinction | Still Existing | Total End of Century |
|---|---|---|---|---|
| IV | 12 | | 0 | 12 |
| V | 12 | −1 | 0 | 23 |
| VI | 9 | −13 | 1 | 19 |
| VII | 1 | −13 | 0 | 7 |
| VIII | 2 | −2 | 0 | 7 |
| IX | 1 | −1 | 0 | 7 |
| X | 1 | −1 | 0 | 7 |
| XI | 14 | | 5 | 21 |
| XII | 22 | −1 | 5 | 42 |
| XIII | 13 | | 11 | 55 |
| XIV | 13 | −1 | 2 | 67 |
| XV | 5 | −11 | 1 | 61 |
| XVI | 18 | −10 | 13 | 69 |
| XVII | 22 | −11 | 11 | 80 |
| XVIII | 7 | −16 | 4 | 71 |
| XIX | 91 | −16 | 91 | 146 |
| XX | 33 | −2 | 33 | 177 |
| | 276 | −99 | 177 | 177 |

Among the general—common—causes of abolition, he mentions not only structural, group-internal, but also incidental, context-dependent, developments.[9]

Finally, if we look at historical studies focusing on the decline of religious life in recent times, we see that even in these, dying and disappearance as such are not central issues. Even Hostie, who perceives (in 1972) an "all-defying stability" (Hostie 1972, p. 260) and a "halo of imperishability" (Hostie 1972, p. 261) in the twentieth century, goes no further than to speak of the phenomena of rigidity ("Furthermore, no one has yet the possibility of becoming aware of how much this stability degenerates into rigidity"[10]) and aging ("One can observe all the signs of aging there. Its influence is dying out, its activities are languishing, its membership is shrinking"[11]). A good historical analysis of the "decline of religious life" in the twentieth century is offered by the theologian Anton Lingier and the sociologist Wim Vandewiele, who take a historical, social, ecclesiastical, and theological perspective. They do mention the death and abolition of the orders, but otherwise remain mainly focused on the phenomenon of decline (Lingier and Vandewiele 2021; See furthermore Sloot 2015). Conversely, to conclude this section, historical studies on the experience of death and the afterlife do not pay attention to organizational dying.[12]

## 3. Sociological Perspective

Very important for understanding how "the period of active organizational involvement may already be coming to an end" (2006) is the work of American sociologist Patricia Wittberg, who has studied the "effect of denominational groups losing their institutional identities" extensively.[13] Ironically, and notwithstanding the groundbreaking quality of her *organizational* research in a sociological perspective, Wittberg does not study the effect on denominational groups of losing their institutional *communities*, viz., their houses and their sisters and brothers. The ending process of "falling" religious orders and congregations that she studies is therefore not a dying process of "closing" religious groups.[14] In other words, should sisters and brothers belonging to those dying groups read sociologist Wittberg's work in order to find some orientation or even consolation, they would to a greater or lesser extent recognize themselves in what she describes as institutional "goal restatements" (still serving others, outside the institutions; personal and spiritual growth) (Wittberg 2006, pp. 173–80) but would find even her most astute observations (e.g., "It is time for them to read these chapters, to assess the story they have told, and to choose, insofar as they

can, the story of their future" (Wittberg 2006, p. 279)) less useful. Observations such as these are no more than half-helpful, because groups of religious that have lost their schools, hospitals, and the like[15] are now losing their houses, their families, and their lives.

Other sociological literature on death is equally unsatisfying in this regard. This becomes apparent when we read sociologist Zygmunt Bauman's *Mortality, Immortality, and Other Life Strategies* from the same viewpoint of the immediate threat of corporate death (Bauman 1992; See furthermore Howarth 2007; Seale 1998; Thompson and Cox 2017). His philosophical exploration of death and dying in social institutions and culture poses relevant questions but offers no real answers to the situation at hand. The remaining religious in dying communities will be challenged and provoked rather than consoled and supported by phrases such as:

- death is the explicit target of many of the things we do and think of (Bauman 1992, p. 7)
- the constant risk of death [ . . . ] is, arguably, the very foundation of culture (Bauman 1992, p. 31)
- the idea of self-preservation [ . . . ] hides or beautifies the gruesome truth of survival (Bauman 1992, p. 34)
- denial of nothingness lacks all solidity if anchored in the future (Bauman 1992, p. 54)
- the terror of death may be exorcised in more than one way (Bauman 1992, p. 173)
- mortality needs not be deconstructed: it ought to be lived (Bauman 1992, p. 191)
- in this readiness for self-sacrifice [ . . . ] the liberation from the tyranny of self-preservation is won. (Bauman 1992, p. 202)

For those who have made spirituality as the creative and caring engagement with humanity their profession (which has come to an end a long time ago) and path of life (which is now coming to an end), reading Bauman is an exercise in cynicism from the very first pages, where we read that

> [ . . . ] the perceiving subject may only delude itself with a play of metaphors, which conceals rather than reveals what is to be perceived, and in the end belies the state of non-perception which death would be. Failing that, the knowing subject must admit its impotence and throw in the towel.[16]

## 4. Organizational Perspective

With regard to the study of the organizational death of denominational groups, three things should be established beforehand. First, organizational death is not (yet) part of the broader field of thanatology, "the description or study of death and dying and the psychological mechanisms of dealing with them."[17] Second, the final phase of existing religious congregations and communities has barely been scrutinized from the scholarly viewpoint of organizational death.[18] Third, existing organizational death research from the field of organization studies and public administration is not, at least not explicitly, directed at religious institutions (See especially, Bell et al. 2014; Bell and Taylor 2011; Sutton 1983; Sutton 1987). When we take a closer look at these publications on organizational death, the amount of which is still relatively modest, we see how organizational death theory can be very useful for our purpose. First, it is useful in offering the right terminology: scholars of organization Emma Bell, Janne Tienari, and Magnus Hansson, for example, speak about cessation of organizational function; mourning and memorialization; and the construction of symbolic immortality—terms all applicable (in various degrees) to our case.[19] Second, it is useful in translating issues related to individual death to collective demise: scholars of organization Emma Bell and Scott Taylor, for example, suggest sound ways of generalizing findings from studies of bereavement at the individual level to organizational contexts.[20] Third, organizational death theory is potentially useful in offering stage and transition theories, one of which is Stuart Albert's model of four psychological "closure-constructing" devices.[21]

The Conference of Dutch Religious (Konferentie Nederlandse Religieuzen[22]) has drawn up a digital document for handling an upcoming phaseout.[23] This "Script for completion" (Draaiboek voltooiing) contains checklists of tips and tricks with regard to seven areas: governance, charism, housing and buildings, member care, finances, Superior's advisors, and staff. The introduction page reads:

> How so completion?
>
> By the term completion we mean that the existence of a religious institution or society of apostolic life in the Netherlands, humanly speaking, is coming to an end.
>
> If you doubt whether your institution is in the phase of completion, consider being twelve years away from now, and then ask yourself the following questions:
>
> Are there members senior enough to run the institute?
>
> Are there still enough members to hold chapters?
>
> (For contemplative monasteries): Are there still at least five members, at least one of whom is under the age of seventy?
>
> If you have to answer no to any of these questions, your monastery is in the stage of completion.[24]

Considering the rapid disappearance of organizational patterns, the document's approach is practical, technical, and juridical.[25] The treatment of the spiritual, experiential, and personal aspects of an unfolding transitus is reduced to a minimum,[26] but whether help with these aspects can be found outside or inside one's own organization remains unclear, at least to the visitor of the website.

## 5. Psychological Perspective

A characteristic, fatal, psychological group mechanism—disconnection from an awareness of the real needs and conditions of the world—is mentioned by Hostie, who argues that active religious groups refuse to languish when the task operation no longer meets the current need: "With such conviction one continues to support the work, and one or more centuries pass before the members of the group become aware that the goal they are pursuing no longer exists. It is then too late to switch to other tasks." (Hostie 1972, p. 127). This might be applicable here as to other historical "stagnations" and "extinctions" (Hostie) of religious orders, but we have to consider that congregations and communities have been quickly overtaken by developments in the decades before and after the Second Vatican Council.[27] Instead of looking backward and trying to answer the question how the sisters and brothers could have responded more effectively to shifting circumstances, it is perhaps more useful for our purpose to look forward and try to answer the question how the members of dying communities can cope with the process of termination or transformation now underway. Not all of them have been as well prepared for private death as the members of some Benedictine abbeys

> where the novice master takes the candidate to the cemetery on the day of his entrance, to show him the place where he will rest after his death. Thus it is made clear to the candidate that from now on he binds his earthly life to the community of the abbey, as a preparation, indeed as an anticipation, of the future life. (Hostie 1972, p. 88)

Nor have they been prepared and trained to die a good death collectively.

The psychological literature on death, dying, and bereavement is vast, its general focus being on psychological, social, spiritual, and medical aspects of individual dying (See especially, Jong and Halberstadt 2016; Kastenbaum 2000; Kübler-Ross 2014; Kübler-Ross 1975). Many authors use the groundbreaking work of Swiss-American psychiatrist Elisabeth Kübler-Ross, whose legendary model of five stages of death (based on interviews with over two hundred dying patients) has remained highly relevant since its publication in

1969.[28] The first stage is that of denial and isolation. Denial "functions as a buffer after unexpected shocking news, allows the patient to collect himself and, with time, mobilize other, less radical defenses." (Kübler-Ross 2014, p. 38). The second stage is that of anger. It may be directed at specific persons or be generalized and undirected and, as a natural response, it requires tolerance and understanding from those around the dying person.[29] The third stage is that of bargaining: "maybe we can succeed in entering into some sort of an agreement which may postpone the inevitable happening." (Kübler-Ross 2014, p. 79). The fourth stage is that of depression, which can be twofold: reactive depression, due to the many losses—money, job, care of children—a patient has to endure, and preparatory depression, "which does not occur as a result of a past loss but is taking into account impending losses." (Kübler-Ross 2014, p. 85). The fifth stage is that of acceptance:

> If a patient has had enough time (i.e., not a sudden, unexpected death) and has been given some help in working through the previously described stages, he will reach a stage during which he is neither depressed not angry about his "fate." He will have been able to express his previous feelings, his envy for the living and the healthy, his anger at those who do not have to face their end so soon. He will have mourned the impeding loss of so any meaningful people and places and he will contemplate his coming end with a certain degree of quiet expectation. (Kübler-Ross 2014, pp. 109–10)

If we substitute the singular with the plural here, and include in our thought experiment Kübler-Ross's constant consideration of communal resources (notably family members), we are already beginning to see how religious sisters and brothers can be brought to that stage of having had enough time [to prepare for death], of having received some help in working through the previous stages, of having been able to express their previous feelings, of having mourned the loss of people and places, of contemplating their coming end with a certain degree of quiet expectation.[30]

## 6. Theological Perspective

In his study on the life and death of religious orders, Hostie somewhat cynically remarks: "At first glance, one may be surprised to encounter such a stubborn desire to stay alive among groups who proclaim that they do not fear death but see it as a transition to a better life."[31] He speaks in this regard about a "dogged defiance of death" once groups of religious have been established and constructed.[32] Once these groups have entered the final stage of life, however, the stubborn desire to stay alive and the dogged defiance of death will at some point have to be relinquished and replaced by the reinforced awareness that "everything earthly is drawn on the sand of transitoriness" (von Balthasar 2012, pp. 15–16; Reference in Levering 2018, p. 3). The transitoriness of earthly existence, often dealt with in relation to salvation, redemption, and divine judgment, is one of most essential biblical truths. In the New Testament, in which the death of Christ is proved to contain for every believer the promise of a better life after this life,[33] the central theme of lasting glory in heaven's kingdom versus passing glory in this life involves and invites reflections on its implications for individual life and death, but several of these reflections may also be interpreted as inspirational for the *collectively* dying and members of death-facing communities.[34]

Beyond the Bible, in Christian theologies of death, which generally focus exclusively on individual dying, other strengthening and soothing strategies for transitioning together from life to death—from death to life—may be found. A good example is *Dying and the Virtues* by theologian Matthew Levering (Levering 2018; See furthermore Davies 2008; Jones 2007; Rahner 1972), in whose reflection on good ways of dying I identify three such strategies. The first strategy is that of fostering virtues. Levering discusses love, hope, faith, penitence, gratitude, solidarity, humility, surrender, and courage as the virtues indispensable for the difficult process of departure and dying. Were these, I would ask, not already virtues indispensable for the demanding process of devotion and dedication of those who were then flourishing and contributing to life but now wilting and withdrawing

from it as a group?[35] The second strategy is that of offering ourselves. Levering states that "[A]ll we can do is to practice offering ourselves to God with the recognition (possible in the *fullest* way only when we are truly dying) that in reality 'one possesses only what one lets go of.'"[36] Was this, obviously, not already the basic premise and practice of those living in religious orders?[37] Furthermore, "The hope that Christians possess in dying—the Lord to whom we give ourselves in dying—is the same Lord to whom we give ourselves in living." (Levering 2018, p. 88). Were the religious, again, not already the exemplary givers of themselves in living? The third strategy, finally, is that of seeking communion. Once more, this poses the obvious question: is this act, which aims at eucharistic communion in Christ, interpersonal communion, and communion with the body of Christ (with those living and dying in the Church as a growing gathering), (Levering 2018, pp. 3, 8, 10) not most natural for those who already live together in unity, "intent upon God in oneness of mind and heart"?[38]

## 7. The Future of Religious Life

Like individual dying for most individual persons, collective dying for most communities will be a brutality, a trial, and an agony (See Levering 2018, pp. 164, 165, 167). Instead of "explaining away the bitter inevitability of their own death," (Levering 2018, p. 165) however, religious men and women should develop their own ways—or adopt those of others—of dying well unitedly.[39] Elements of such *ars conmoriendi* would include soothing strategies such as the above-mentioned but also, among others, shared eschatologies (orientations on "ends" instead of "evolutions"[40]) and suitable symbols (reflections on images of death and fruition[41]). As for the near future, many religious congregations are faced with their fall (Wittberg 1994) and thus with the critical choice between cessation (termination) and continuation in some new form (transformation). Both options can be viewed by those involved as (extremely) negative or positive.[42] The case of the Franciscan Sisters of Veghel in the Netherlands shows that, despite the vanishment of much vitality, the execution of the first option may not be as simple as it seems (See Pansters 2023, forthcoming). Whereas the remaining sisters are moving into a new housing facility just behind the old monastery,[43] their monumental complex ("Kloosterkwartier") will soon be housing a dozen welfare organizations.[44] The negotiations between the sisters, who primarily want to pass on their spiritual values, and the organizations, which are more entrepreneurial and target-oriented, have been difficult at times. At any rate, the Dutch branch of the congregation, founded in 1844, will come to an end at some point in the near future. The case of the Franciscan Brothers of Huybergen in the Netherlands, who have decided to keep their monastery and revitalize their community by inviting a spiritual group of laypeople into their midst, reveals –not unsolvable – problems of another nature: the traditional values of an age-old teaching congregation are not easily combined and reconciled with the green goals of a "place with attention for earth and breath."[45] Viable avenues are now being formulated. Similar cases in the Netherlands and Germany show that the members of religious congregations are more easily reconciled with their fate when they see that their spiritual heritage, which includes their buildings and gardens and graveyards, is not being reduced to some matter-of-fact, theologically decontextualized, "material heritage."[46]

## 8. Conclusions

For over half a century, traditional forms of organized religious life have been disappearing from the scene in Western society. The dying process and/or death of orders and congregations is probably intensely painful for the communities involved. Whereas a small minority of their members may rather combatively try to counter institutional decline and decay with all creative means possible, most show themselves resigned and reconciliated with fate—"Thy will be done." Insecurity and equanimity exist side by side. Subjective accounts of the challenge of letting go and saying goodbye may be very rare, but we can safely assume that the assault on psychological powers is extremely discouraging and

draining on personal and communal levels. Of equal importance is the loss of religious resources—spiritual ideas, rationalities, moralities—on a cultural level. For religious leaders, the crux of the question therefore remains: retreat or renew, terminate or transform, seek death or life?[47]

For both these internal, spiritual, and external, social, reasons, I contend that we need a new theology of religious life and death. This theology will have to address "all" death-related issues presently experienced by counts of religious men and women, including spiritual (devotion, desire, destiny, and deliverance), psychological (difficulty, distress, denial, and defiance), and material (decay, decline, discontinuity, and demolition) ones. As an extensive reflection on the multiple meanings of transitus, involving variations of termination and transformation, this theology would tackle matters of ultimate concern (including those of traditional concern such as the four last things of death, judgement, heaven, and hell) and thus also reveal its relevance for current concerns in contemporary culture, where—as of in one parallel movement with the marginalization of religious living—life is lived only once and death is marginalized and ignored. (Davies 2008, pp. 10, 20, 171, 178). For the religious themselves, whose sense of singleness of purpose of life (Davies 2008, p. 71; See also Pansters 2020) rarely spills over into a shared strategy or even vision of death, the exhausting experience of dying together oscillates between hope and fear, between embrace and repudiation. Ultimately, their death struggle is concomitant with the tension between creation (life) and salvation (death) that is the core dynamic of the Christian commitment. (See Davies 2008, pp. 5, 7, 9, 41, 57, 120).

> *Trust*
>
> Making the world more beautiful together
>
> getting into a more positive flow
>
> looking for something nice and positive
>
> a small beautiful deed, something sweet
>
> my faith, my intense experience
>
> in a more beautiful environment
>
> I started to see myself
>
> what can I do maybe
>
> slowly I started to build
>
> started trusting myself
>
> Enjoying the love around me
>
> I believe in my dream that appeared
>
> There I stand
>
> I know I can do it
>
> I have made the decision
>
> To do what I can dream
>
> my fears I get rid of
>
> trust is what leads me
>
> With strength and faith truly choose
>
> Or else a dream and time to lose
>
> should I hesitate and talk for days
>
> my certainties I will now leave behind
>
> people will call me crazy
>
> And from others I will experience love
>
> Realizing dreams takes guts
>
> that is what I realize inside

Letting go is what really needs to be done

I feel it will surely come right

For I believe that if I really do fall

there will always be someone who will

catch me.[48]

**Funding:** This research received no external funding.

**Informed Consent Statement:** Informed consent was obtained from all subjects involved in the study.

**Acknowledgments:** I would like to thank David Couturier OFM Cap., Willem Marie Speelman, and Wim Vandewiele for their valuable help and feedback.

**Conflicts of Interest:** The author declares no conflict of interest.

## Notes

1  See for example (McLeod 2007). That institutional religion is presently not simply coming to an end in Western societies is argued by Kees de Groot (2018).

2  Examples of post-Vatican II responses pleading for the renewal of religious life are Arbuckle (1988); Azevedo (1988); Cada et al. (1979); Nygren and Ukeritis (1993). These works concentrate on change in times of crisis from an anthropological (Arbuckle), theological (Azevedo, Cada), or psychological (Nygren and Ukeritis) perspective. The latter authors, for example, claim that "For religious orders to continue as a vital force in the Catholic Church, in the United States, and in the world, they must change in dramatic and significant ways. [ . . . ] Since the Second Vatican Council, the changes in the Roman Catholic Church in the United States have effected the way members of religious orders live and work more thoroughly than any other single population" (cover blurb). For an attempt (psychological) to revive the refounding religious life movement, see Dunne (2009). For a Central and Eastern European, sociological and transformational, perspective, see Palmisano et al. (2021).

3  On the different rules of the various religious orders, see Pansters (2020).

4  Wim Vandewiele studies the current dynamics and problems of religious orders and congregations with the help of a (spiritual, social, cultural, physical, financial) "capital" toolbox, starting from Pierre Bourdieu's model but also from the orders' and congregations' *spiritual* capital. This is a sociological frame of reference that might also be interesting to look at from a historical perspective. See Vandewiele (2022).

5  The term "transitus" typically refers to the death of Francis of Assisi. For its Franciscan origin, see Hülsbusch (1974); Ménard (1974).

6  A good example of "particular" cases of organizational death is offered by Roest (2013). In the context of reform, he also speaks of the disappearance and destruction of communities, for example, because of plague or war (pp. 165–66). His concept of "disorder" is also very relevant for our project.

7  See mainly his pioneer volume *Institutionen und Geschichte: Theoretische Aspekte und mittelalterliche Befunde* (Melville 1992) and the series founded by him, Vita Regularis.

8  (Hostie 1972, p. 79). See furthermore: "Do religious institutions have an age? Are they aging? And, if so, is their old age crowned with death? If that is their destiny, then their end is not necessarily an accident of chance, caused by an unexpected confluence of circumstances or by the malicious intent of malefactors. It could also be the normal end point of their own history" (p. 9).

9  For example, "invasions or plunder; wars or epidemics; arbitrary political measures or papal intervention" (p. 89).

10  (Hostie 1972, p. 274). Hostie underlines the crucial role of the task set by the Second Vatican Council: "Everything must be recast, transformed and renewed. And this after nearly a century of doing nothing but stabilizing and maintaining what had been thought of as the definitive and unchanging restoration" (p. 275).

11  (Hostie 1972, p. 315). Hostie also points here to the process of "museumization": "All that remains to it [the religious group] are its buildings, the carefully preserved traces of a great past. It does not like to get rid of them. As long as it is possible, it keeps them, willing as it is to devote all its energy to maintaining and sometimes even beautifying them. The survivors fuse with the stone or walk around in it like the guardians of a museum" (p. 315).

12  See for the Middle Ages, for example, Booth and Tingle (2021); Bynum (1998); Classen (2016); Rollo-Koster (2017).

13  Wittberg (2006), back cover blurb. See especially her *The Rise and Fall of Catholic Religious Orders: A Social Movement Perspective* (Wittberg 1994).

14  Where Wittberg refers to organizational demise, it is in terms of "possibility." For example, "[ . . . ] there is almost no documentary evidence that the possibility of organizational demise was a seriously considered reality that affected either the policies of the congregations or the daily lives of the sisters" (Wittberg 1994, p. 220). In one instance, she does mention "an already dying institution" (p. 266). In a similar way, the authors in *The Transformation of Religious Orders in Central and Eastern Europe: Sociological Insights* (ed. Palmisano, Jonveaux, and Jewdokimow) avoid the discussion of death and zero in on "changes" (Jewdokimow), the

15　search for identity (Jonveaux and Sadlon), transformation (Mirek), routinization (Jonveaux), reinvention (Spalová, Liška and Picková), new communities (Palmisano), presence (Révay), and pilgrimage (Medvedeva) (Palmisano et al. 2021).

15　Institutions that "were an essential and unquestioned component of the *virtuoso* identity of the sisters, deaconesses, and mission society members" (Wittberg 2006, p. 259).

16　(Bauman 1992, p. 2). The important aspect of finding closure (dealing with death) in the case of corporate closure (organizational death) is equally absent from the other sociological books mentioned.

17　https://www.britannica.com/science/thanatology (accessed on 30 October 2022). One of many examples showing that thanatology is very much an interdisciplinary study (which might be receptive to organizational approaches) is King's University College at Western University Canada, which has had a Death Education program since 1976: "The Thanatology department offers a wide variety of courses, including an overview of bereavement and grief, ethical issues, palliative care, suicide, children and death, spiritual and philosophical issues, change and transition, popular culture, grief and trauma, and diversity and social justice," https://www.kings.uwo.ca/academics/thanatology/ (accessed on 30 October 2022).

18　This perspective is also lacking from much general research on the organization of religious institutions, such as Demerath et al. (1998). See Sutton (1987): "Yet only one of the ten life cycle models reviewed by Quinn and Cameron (1983) included a decline phase. Whetten (1987) concluded in his review that organizational death is probably the least studied of any organizational growth or decline process" (p. 542). This still seems to be the case.

19　Bell et al. (2014). Other formulations used (subjects addressed) are corporate closure or shutdown; organizational mortality, discontinuity, and decline; crisis, failure, and collapse; the removal of fundamental structures of work-related meaning; organizational demise; organizational reanimation; and the spiritual, ethical, and embodied dimensions of organizational death.

20　Bell and Taylor (2011, p. 1): "Theories of individual bereavement have thereby acquired the potential to inform understandings of loss and grief at the collective level." The same goes for *feelings* of loss. See, for example, Cunningham (1997): "those who are affected by an organization's death—associated organizations, clients, and members—experience many of the same feelings as when people die" (p. 474).

21　Albert (1984, p. 172). As mentioned and explained by Bell and Taylor (2011, p. 3), "a summary process in which important aspects of the past are evoked and reviewed; a process of justification when reasons for termination are stated and defined; a continuity process, where a link is constructed between past and future; and a fourth process involving 'a momentary increase in attachment . . . akin to a eulogy . . . in which the value of that which will be lost is celebrated in order to create the possibility of closure.'".

22　As an umbrella organization, the KNR is very well established in the Netherlands. There are KNR staff working on the matter of completion, which—unfortunately—is not the case everywhere. In the US, this kind of support for religious congregations and communities seems to be more developed.

23　KNR Draaiboek voltooiing (n.d.). See the explanatory page: "The KNR has prepared this Script for Completion for the benefit of superiors and other directors and administrators of religious institutions and societies of apostolic life in the Netherlands. At a time when so many institutions are seeing the end of their existence coming in the Netherlands, the demand has arisen for a handbook that can help superiors, their councils, and their advisors to responsibly fulfill their task in the final phase" (translation KP).

24　Translation KP. These questions have been based on Wijlens (2017).

25　See, for example, the page on member care: "An important aspect of our lives is caring for the elderly and sick. There are all kinds of intangible aspects and values attached to care. [ . . . ] Daily reality, however, is primarily a matter of practicality. It is inevitable to talk here mainly about material things" (translation KP).

26　See, for example, the page on charism, which gives a description of the importance of community life, celebration and remembrance, apostolate, and pastoral care, but only very minimally on handling change or crisis in these areas.

27　As Dutch historians Annelies van Heijst, Marjet Derks, and Marit Monteiro show with regard to the Netherlands, a "paradox of thriving decline" started already before the Second World War. The religious turnabout after the War continued this paradox in the form of continued renewal during decline. van Heijst et al. (2010, pp. 261–314, 711–35, 799–823, 910–25, 935–1050).

28　(Kübler-Ross 2014, pp. 37–132). Her work is also mentioned in Bell and Taylor (2011, pp. 2–4) and Cunningham (1997, *passim*). The model has been increasingly criticized and many alternative models have been developed in recent years. See Tyrrell et al. (2022).

29　My formulation here is based on Tyrrell et al. (2022).

30　In the context of the current reality of "the challenge of facing the future with, in one sense, 'no future'" and for the purpose of a "spirituality of refounding," pastoral psychologist David Couturier in 1990 asked the following question: "despite our hesitancy to admit to the reality of death, have religious experienced significant losses which amount to an experience of death and dying?" He then explored three types of losses: emotional, spiritual, and communal. Couturier (1990, pp. 81–82).

31　(Hostie 1972, p. 88). David Couturier compares the organizational death of religious congregations to Jesus and his "hour" in John's Gospel and what John is saying to his community around the year 100AD about their impending organizational death. Very few understand what John is doing in asking his community not to resist the hour of their communal death, to take it up as Jesus did, to the glory of God the Father. See Couturier (2008, pp. 123–26).



32    (Hostie 1972, p. 88). See in this regard Becker (1973), a cultural anthropological study exploring how most human action is taken to avoid the inevitability of death.

33    See, for example, "The one who believes in me will live, even though they die; and whoever lives by believing in me will never die" (John 11:25–26); "If we live, we live for the Lord; and if we die, we die for the Lord. So, whether we live or die, we belong to the Lord" (Rom. 14:8); "For to me, to live is Christ and to die is gain" (Phil. 1:21); "And so we will be with the Lord forever" (1 Thess. 4:17). Some inspirational passages from the Old Testament are the following: "Even though I walk through the valley of the shadow of death, I will fear no evil, for you are with me" (Ps. 23:4); "Surely your goodness and love will follow me all the days of my life, and I will dwell in the house of the Lord forever" (Ps. 23:6); "In the way of righteousness there is life; along that path is immortality" (Prov. 12:28).

34    See, for example, "Heaven and earth will pass away, but my words will never pass away" (Luke 21:33); "For this world in its present form is passing away" (1 Cor. 7:31); "And if what was transitory came with glory, how much greater is the glory of that which lasts!" (2 Cor. 3:11); "The world and its desires pass away, but whoever does the will of God lives forever" (1 John 2:17). An inspirational passage from the Old Testament is the following: "There is a time for everything, and a season for every activity under the heavens: a time to be born and a time to die" (Eccl. 3,1–2).

35    I am referring here to what Douglas Davies, in *The Theology of Death*, calls the "interplay," "mirroring," "dialectic," "conjunction," "continuum," "coherence," "match," "alignment," "relation," "resonance," "bond," and "pairing" of "life-style and death-style" (2–3, 18, 38, 54, 57, 61, 98, 118, 148, 151).

36    (Levering 2018, p. 36). Levering is quoting Pieper (2000, p. 92).

37    This with obvious liturgical and christological connotations: "The virtues of dying are those that enable us to exercise this priestly offering, as dying members of Christ's body" (Levering 2018, p. 168).

38    See the *Rule* of Augustine, chapter 1, quoting Acts 4:32, transl. Robert Russell, based on Verheijen (1967), https://faculty.georgetown.edu/jod/augustine/ruleaug.html (accessed on 30 October 2022).

39    To date, there is no *ars moriendi* for organizations. On the individual art of dying well, see, for example, Dugdale (2017); Leget (2007); Marr (2010). For the medieval treatise, see Anonymous (1997).

40    See for example (Metz 2014, pp. 67–68): "Bieten wir Christen aber der Welt nicht das peinliche Schauspiel von Menschen, die zwar von Hoffnung reden, aber eigentlich nichts mehr erwarten? Ist das christliche Leben noch mit zeitlich orientierter Erwartung und Sehnsucht aufgeladen? Blicken die Christen—auch die Ordenschristen!—wirklich noch gespannt auf das Ende? Erwarten sie überhaupt noch ein Ende—nicht nur für sich selbst in der Katastrophensituation des individuellen Todes, sondern für die Welt und deren Zeit? Ist eine Begrenzung, ein Ende der Zeit überhaupt noch denkbar—oder wurde die Erwartung eines Endes der Zeit nicht längst ins Reich der Mythologie abgewiesen, weil die Zeit selbst zu einem homogenen, überraschungsfreien Kontinuum, zur schlechten Unendlichkeit, zu einer leeren, evolutionär zerdehnten und zersetzten 'Ewigkeit' geworden ist, in der alles und jedes passieren kann, nur dies eine nicht: dass nämlich eine Sekunde 'zu der Pforte wird, durch die der Messias in die Geschichte tritt [W. Benjamin]' und in der es *deshalb* Zeit wird für die Zeit?".

41    See, for example, Zweerman (2001, p. 66): "What is certain is that this hoped-for fertility will not take its course without what in the great spiritual tradition is called 'dying' or 'mortification.' This again by analogy with what a fertilized organism shows in nature: relinquishment of a way of flowering, in order to make room for a new life form that can take on a life of its own: the fruit" (translation KP). To the death-related theme of fruition belongs the life-related image of the tree, in particular the tree of life, "whose fruit would feed the faithful" (Davies 2008, p. 26).

42    An unpublished survey among the members of six Franciscan communities in the Netherlands and Belgium revealed a broad spectrum of perceptions of the collective future, from very pessimistic to very optimistic. See Krijn Pansters and Willem Marie Speelman, "Verslag gesprekken met vertegenwoordigers communiteiten" (KNR Project "Transitus: Franciscaanse spiritualiteit tussen eindigheid en het uiteindelijke") (unpublished).

43    Zusters Franciscanessen van de Onbevlekte Ontvangenis SFIC, http://www.sficnet.org/index.php/nl/ (accessed on 6 December 2022).

44    Leefgoed Veghel, https://leefgoedveghel.nl/ (accessed on 6 December 2022): "Based on the ideas of the Franciscan Sisters, together they make the Kloosterkwartier in Veghel a place for everyone: young, old, rich, poor, with and without disabilities. You are already welcome in our temporary Meeting Center. In time there will be more activities in the area of 'Learning, working and experiencing together' and 'Caring and experiencing together'" (translation KP).

45    Broeders van Huijbergen, https://www.broedersvanhuijbergen.nl/ (accessed on 6 December 2022); De Huijberg, https://dehuijberg.nl/ (accessed on 6 December 2022).

46    "Similar cases": a two-year pilot project that studies "transitus" in five religious communities in the Netherlands is now underway. We combine the examination of historical and organizational documents with interviews with stakeholders as well as inspirational meetings. The results will be published in 2024. "Spiritual heritage": some secular organizations are cultivating the spiritual heritage, for example, Kloosterkracht, https://kloosterkracht.nl (accessed on 6 December 2022); Networking Intentional Christian Communities, https://www.nicc.network (accessed on 6 December 2022); Vereniging Religieuze Leefgemeenschappen, https://verenigingreligieuzeleefgemeenschappen.nl (accessed on 6 December 2022). "Material heritage": some organizations are focusing on the material heritage of former monasteries and churches, for example, Klosterland e.V., https://klosterland.de

(accessed on 6 December 2022); Transara Sakralraumtransformation, https://www.transara.uni-bonn.de (accessed on 6 December 2022); Wissensportal Transformation von Klösters, https://zukunftkulturraumkloster.de (accessed on 6 December 2022).

47     In the Netherlands, a publication entitled *Sterven of werven?* (*Die or Recruit?*) appeared already in 1987. In the foreword we read the following: "So is this book a death message in disguise or a restorative approach that seeks to squeeze new wine into old bags? The answer to both questions is no. To the extent that conclusions are drawn by the author, they do not testify to a desire for death or for the restoration of old certainties. In his exploratory research, he continually and persistently encounters boundaries. Institutional boundaries of his own congregation, of the Church and of the Church Code. But also limits of individual people who who do not all have the same radical and uncompromising way to follow Jesus Christ. As the end of the book draws near, the notion of 'recruit or die' becomes less and less important. The author leaves these words behind, having turned them inside out before the eyes of the reader. What he wants to focus on is the future and life. And these lie beyond the confines of one's own circle: 'Let us forget the boundaries of codex and congregation and seek life beyond them.'" (Sponselee 1987, p. 8) (translation KP).

48     Poem by Mark Verhees in the "Strategische Meerjarenplanning in de fase van voltooiing van de organisatie rondom de Nederlandse communiteit van de zusters Franciscanessen van Veghel" (11 June 2019); internal document of the Franciscan Sisters of the Immaculate Conception of the Holy Mother of God.

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
