# Peer review of "Transitus: The Discontinuity and Death of Religious Communities in the Twenty-First Century"

_religions, doi:10.3390/rel14030354_

Round 1
Reviewer 1 Report
The topic of the reviewed article is not new in the monastic literature. However, what is relevant here is the presentation of the different perspective on the topic. I would recommend minor revisions:
1/ Translating table 1.
2/ In the chapter on sociological perspective please add other sociological texts that deal with the transformations of the religious orders. One can find them in S. Palmisano, I. Jonveaux, M. Jewdokimow 2021. Tranformations of the Religious Orders... and M. Jewdokimow, 2021 A Monastery in a Sociologial Perspective... Following the second book the Author will also find out that P. Wittberg (quoted in the subchapter on the sociological perspective) presents in fact the organizational approach in the sociological sociological perspective.
3/ Hence, please revise the difference and content between the organizational and sociological perspectives subchapters.
4/ I would like to read more about the problem under scrutiny in the Netherlands and Belgium. I suggest to make it more a case study research and provide more data that would address the topic - for now the Author provides only selected information.
Good luck with your revisions!
Author Response
Dear Reviewer,
Thank you very much for your kind and helpful response. I have improved (in red) the points that you mentioned:
1 I have given a translation of the table in footnote 15.
2 I have added the study of Palmisano et al. that you mention. Thank you for directing my attention to it. It made me aware that I had left out the Central and Eastern European perspective and I am glad that I could include it here. I have now added it in footnote 4, where I also added the different scholarly perspectives (anthropological, psychological, etc.) of the other current literature. I have called this study ‘sociological and transformational.” In footnote 24, I have again mentioned Palmisano et al. as important for the sociological perspective next to Wittberg. I have also mentioned here that this volume focuses on transformation (search for identity, reinvention, etc.) rather than organizational death.
To emphasize Wittberg’s organizational approach within the sociological perspective, I have now emphasized “the groundbreaking quality of her organizational research in a sociological perspective” in the text (at 132) and again, a few sentences later, “read sociologist Wittberg’s…” (at 137). This to make sure that the reader understands that Wittberg is a sociologist, nothwithstanding her explicit organizational interest, as can be read at her department’s website (https://raac.iupui.edu/about/who-we-are/research-fellows/patricia-wittberg/) and in her two main books (e.g., on Amazon).
3 It was difficult for me to move Wittberg to the organizational perspective, as you suggested. It was a good and fitting suggestion, except that it would mess with the structure of the article, in which I have dealt with (only) two cases per perspective. In the organizational perspective, I already have the important cases of 1) Bell and co on the term “organizational death” and 2) the Dutch KNR’s “Script for completion”. The latter two cases make clear what I mean by – exclusively – organizational. My extra explanation with regard to Wittberg in the sociological perspective, on the other hand, makes clearer to the reader now why I should deal with her in the sociological perspective, although she is studying organizations. Furthermore, the sociological perspective of Palisano et al. that I have now added as a kind of comparison with Wittberg strengthens the sociological paragraph. I hope you can agree with this estimation.
4 I would have loved to give the reader more insight into the Dutch and Belgian situation from an empirical perspective, but the fact is that our project on “transitus” is the very first in the Netherlands (and probably worldwide) and it is still underway, with results expected in 2024. So I cannot give more information besides the two examples (Veghel and Huijbergen) that I have described in general terms. It is a pleasure to read that you, reviewer, have a real interest in this and it confirms that our current work in the field (among religious communities in the Netherlands) is worthwile.
Reviewer 2 Report
The subject of the "organizational death" of religious congregations is a much-needed but often neglected area of research. As stated in the article, previous research has been restricted to theories of decline not death. The author opens up a new avenue of study with skill and insight. More attention might be given to the substantial research done more recently in the David Nygren and Miriam Ukeritis major study on The Future of Religious Orders in the US (FORUS). It is more current and scientific than the Hostie study. But, this is a minor cavil in an otherwise sturdy introduction to research that this paper demonstrates. There is more research to be done and the author has provided incentive to other authors to match his prodigious beginning.
Author Response
Dear Reviewer,
Thank you very much for your kind and helpful response. I have emphasized the study of David Nygren and Miriam Ukeritis by explaining its (cl)aim in footnote 4. Your remark was important for me to see that the studies that I mention in this early footnote as examples of research that does not acknowledge the problem of death, have particular perspectives themselves: anthropological, psychological, and so forth. I have made these explicit in this footnote.